## [Peer Review File · Nature Communications]

Reviewers' comments:

Reviewer #1 (Remarks to the Author):

In the paper titled "Dynamics of a qubit while simultaneously monitoring its relaxation and dephasing", the authors present a result that is an extension of previous work (References 10 and 12). In Ref 12 two incompatible dispersive readout channels are monitored simultaneously, and in Ref 10 the fluorescence of a qubit is monitored. In the current work, the monitoring of a single dispersive channel and a fluorescence channel are monitored simultaneously. The authors split the output of a cavity, containing a single transmon qubit, to two parametric amplifiers, allowing them simultaneously and continuously measure the relaxation and dephasing channels of the qubit. The authors present the analysis of the quantum trajectories, and show a good agreement with theory. The manuscript is well written. The analysis is thorough and detailed, and the authors have made an effort to make as much of the data as possible available to the readers.

This work is new, yet it is an incremental change compared with Refs 10 and 12. I do think that the results may be of interest to others in the circuit-QED field, yet I would like to see the authors further clarify the distinction between Refs 10, 12 and the current work. The authors have touched up on this towards the end of the paper.

An additional concern of mine is the low quantum efficiency of the fluorescence channel. The low quantum efficiency in the experiment makes the impact of this channel on the results very small, yet it constitutes a main part of the claims of the paper. The effect can be seen in Fig.5, where the fluorescence data shows almost a mixed state, and the combined dispersive and fluorescence are not very different from the dispersive measurement only. It would have been a lot more impressive if the quantum efficiency of the fluorescence channel was significant. If it is not possible to get a higher quantum efficiency in experiment, simulated data in the supmat would be helpful. The authors should also discuss this point to make it clear to the reader.

Before I can recommend the paper for publication I would like the authors to address the issues above as well as the minor comments below:

1. In the abstract the authors write "...we retrieve a significant part of the lost information in both of these channels." I think significant is too strong of a word here, with regards to the fluorescence channel.
2. How much of the qubit signal from the decay reaches the detector vs internal cavity losses vs losses in the path to the detector? This should be in the main text, and not left for inferring from the supmat. On a related note, Is the T1 qubit decay Purcell limited?
3. In Fig.1a, it is difficult to see where the trajectory is going, where does it start/end? How far into the Y plane does it enter?. Can't it be drawn nicer maybe like in Fig. 2a?
4. Where the authors discuss the tomography:
 - a.They show it can be done by direct averaging. This cannot be done in Ref 12, and adds interest to the current result. I think the authors should mention this.
 - b.How does this compare to doing regular tomography (projective measurements with the dispersive readout channel)? To get the same error-bars for a certain state, do you need more traces than you would projective measurements?

Reviewer #2 (Remarks to the Author):

The manuscript by Huard and collaborators realizes for the first time a superconducting qubit continuously measured by both a dephasing measurement and a dissipative measurement, showing the conditional and unconditional dynamics that results. It is only recently that such direct investigations of continuous measurements on superconducting qubits have become possible, and it opens up a new area of research into the measured dynamics of individual quantum systems. This experiment is on the cutting edge of this endeavor, and as such I think the manuscript is appropriate for nature communications.

Reviewer #3 (Remarks to the Author):

The authors report experimental results on the dynamics of a driven qubit under simultaneous monitoring of its relaxation and dephasing. A superconducting qubit in a cavity is subject to continuous resonant drive, while the spontaneous emission field from the qubit is continuously measured with a heterodyne measurement, giving the information about x and y components of the qubit state. The z component of the qubit state is monitored also continuously with a dispersive measurement via the homodyne measurement of the cavity transmission. By changing the amplitudes of the qubit drive and the cavity probe, the authors investigated different regimes of the qubit dynamics both in the underdamped Rabi oscillation regime and the strongly dephased Zeno regime.

The authors took millions of the time-dependent signals for each drive and measurement conditions. By directly averaging the data, they demonstrated the evolution of the qubit state as expected from the master equation. Based on stochastic master equation, they also obtained individual quantum trajectories for a given set of measurement records. The distributions of the qubit states conditioned on the measurement records vividly show how the quantum state evolves under the back-actions from the different detectors.

This is the first demonstration of the simultaneous monitoring of all the three components of the qubit state vector, showing the beautiful "textbook" results. Seeing is believing. The experiment was conducted with very careful calibration and analysis detailed in the Supplementary Materials. I recommend publication of the manuscript in Nature Communications as it is.

As minor comments I would like to point out a few things in the following.

1. The videos in the Supplementary Materials are wonderful. It would be nice to have a table of the 30 representative configurations somewhere so that the readers can easily choose a particular one without waiting for downloading.
2. For the applications discussed in the conclusion, are the current values of the efficiencies η_f and η_d enough?
3. In Fig. S8b, the direction the arrows are aiming at does not obviously look to be $\sigma_z = -1$. How should I interpret it?

Detailed reply to the referees:

First, we would like to thank the referees for taking the time to read our manuscript so carefully and for their detailed comments.

Replies to Referee 1

Referee: This work is new, yet it is an incremental change compared with Refs 10 and 12. I do think that the results may be of interest to others in the circuit-QED field, yet I would like to see the authors further clarify the distinction between Refs 10, 12 and the current work. The authors have touched up on this towards the end of the paper.

Reply:

The main difference between our work and any previous work is that we measure measurement records whose average value correspond to the three directions of the Bloch sphere. Any previous work was limited to two records at maximum.

On top of that, there are other distinctions that we can draw with the two previous works having measured 2 records simultaneously:

- In Ref.[12], by Hacothen-Gourgy et al., they use a dynamical qubit instead of a real one, in the sense that the qubit is continuously driven at its transition frequency. Besides, their measurement corresponds to effective measurement of Pauli operators like our dispersive measurement while our fluorescence channel adds a measurement sensitive to the lowering operator.
- In Ref.[10], by our group, we used two quadratures of a single decoherence channel (relaxation). The average action of each of them is thus identical despite them having a different measurement backaction.

In the manuscript, we had already mentioned some of these points but almost exclusively discussed the distinction with [12]:

p. 1 “therefore extending the previously explored case of two incompatible measurement outcomes~\cite{Campagne-Ibarcq2016,Hacothen-Gourgy2016} to the case of three spin directions.”

p. 4 “which is similar to what is obtained by simultaneously measuring σ_x and σ_z in an effectively undriven qubit~\cite{Hacothen-Gourgy2016}.”

p. 5 “This asymmetry highlights the profound difference between measuring both quadratures of fluorescence and measuring σ_x and σ_y simultaneously using dynamical states as in Ref.~\cite{Hacothen-Gourgy2016,Vool2016}. While both methods lead to the same result on average, their backaction differs. The latter corresponds to quantum non-demolition measurements, while fluorescence does not.”

As suggested by the referee, we add in page 3 the following distinction to [10]

“Interestingly, the two fluorescence measurement records σ_u and σ_v exert a different backaction but act identically on average (same Lindblad operators). The additional

dispersive measurement that we introduced compared to Ref.~\cite{Campagne-Ibarcq2016} thus leads to a very different dynamics.”

And following the point 4a below, we add on page 2

“Such a direct full tomography cannot be done by measuring two records only~\cite{Campagne-Ibarcq,Hacohen-Gourgy2016}.”

Referee: An additional concern of mine is the low quantum efficiency of the fluorescence channel. The low quantum efficiency in the experiment makes the impact of this channel on the results very small, yet it constitutes a main part of the claims of the paper. The effect can be seen in Fig.5, where the fluorescence data shows almost a mixed state, and the combined dispersive and fluorescence are not very different from the dispersive measurement only. It would have been a lot more impressive if the quantum efficiency of the fluorescence channel was significant. If it is not possible to get a higher quantum efficiency in experiment, simulated data in the supmat would be helpful. The authors should also discuss this point to make it clear to the reader.

Reply:

As the referee points out, the effects we observe would have been more dramatic had we managed to reach higher detection efficiencies. However, the distortion of the trajectory statistics when fluorescence records are also taken into account is already clear and insightful in Fig. 5. In particular:

- the qubit state y coordinates are now spread instead of being exactly zero
- there is an asymmetry between the spread along y and x when the qubit state is close to $z = 1$ compared to $z = -1$, which is directly related to the fact that $z = -1$ is a pointer state of the fluorescence channel

The effects we discuss do not develop when the efficiency exceeds a particular threshold and therefore, the impact is purely a matter of contrast in the Bloch sphere. For efficiency to truly reveal new insight, one would need to get very close to unit efficiency so that one could restrict the dynamics to pure states only. Yet we all still have work to do in this direction since the best efficiencies are close to 50% and therefore still far from a case where one can consider trajectories on the sphere surface only.

The referee makes an excellent suggestion to simulate what an experiment with unit efficiency could reveal and we now show such a simulation in the supplemental material in Fig. S9. There, one can see on efficiencies ranging from our experimental situation to 100% that our experiment qualitatively matches the statistics of trajectories up to 75% very well.

Referee: 1. In the abstract the authors write “..we retrieve a significant part of the lost information in both of these channels.” I think significant is too strong of a word here, with regards to the fluorescence channel.

Reply:

We agree and have removed this adjective.

Referee: 2. How much of the qubit signal from the decay reaches the detector vs

internal cavity losses vs losses in the path to the detector? This should be in the main text, and not left for inferring from the supmat. On a related note, Is the T1 qubit decay Purcell limited?

Reply:

As far as we know, there is no known technique to differentiate in situ the contributions of each imperfection in the total fluorescence efficiency. We can however estimate theoretically what would be the Purcell rate of the transmon by using a numerical electromagnetic simulator or even cruder expressions if we use simple electromagnetic models. We had designed the qubit to be Purcell limited indeed. However, as we have shown in the thesis of Philippe Campagne-Ibarcq, these expressions fail to reproduce the experimental values of relaxation rates (sometimes, the predicted Purcell rate exceeds the total measured relaxation rate!). Therefore, we do not believe it makes sense to give a precise estimated value for the Purcell rate and do not want to make an unjustified claim on the exact dominant origin of the limitation in efficiency.

What we can say for sure is that the insertion loss between the qubit and the first amplifier of the fluorescence channel is at least 1 dB given the specifications of the components that are on the way.

Referee: 3. In Fig.1a, it is difficult to see where the trajectory is going, where does it start/end? How far into the Y plane does it enter?. Can't it be drawn nicer maybe like in Fig. 2a?

Reply:

Indeed, it is hard to render this trajectory clearly. We wanted to keep it looking as a cartoon in order to convey the idea of the experiment without disturbing the reader by details at this stage. In this new version, we have tried to improve the Figure by adding arrows on the trajectory in order to indicate the arrow of time. We have also improved the colors. However, we did not find a good way to represent all the coordinates of the Bloch sphere without losing in clarity of the message by projecting the sphere on three planes as in Fig. 2.

Referee: 4. Where the authors discuss the tomography: a.They show it can be done by direct averaging. This cannot be done in Ref 12, and adds interest to the current result. I think the authors should mention this.

Reply:

We thank the referee for his suggestion. We now mention it explicitly.

p.2 "Such a direct full tomography cannot be done by measuring two records only~\cite{Campagne-Ibarcq,Hacohen-Gourgy2016}."

Referee:b..How does this compare to doing regular tomography (projective measurements with the dispersive readout channel)? To get the same error-bars for a certain state, do you need more traces than you would projective measurements?

Reply:

We had dedicated a whole section in the supplemental material to this interesting question (section I.D.)

We now refer to it explicitly in the main text on page 2:

“A comparison of between this new technique and the usual technique using a qubit rotation followed by a projective measurement is discussed in the supplemental material\cite{supmat}.”

Replies to Referee 2

We thank the referee for his positive assessment of our work.

Replies to Referee 3

Referee: As minor comments I would like to point out a few things in the following.

1. The videos in the Supplementary Materials are wonderful. It would be nice to have a table of the 30 representative configurations somewhere so that the readers can easily choose a particular one without waiting for downloading.

Reply:

We are glad the referee appreciated the videos. We have followed the above advice and realized a table illustrating the 30 different configurations in the supplemental material. In this new table Fig. S2, each of the panel hides a hyperlink to the corresponding video.

2. For the applications discussed in the conclusion, are the current values of the efficiencies η_f and η_d enough?

Reply:

Indeed, we believe that all the applications we mention (statistical properties of quantum trajectories, fluctuation relations in quantum thermodynamics, quantum smoothing protocols, and to parameter estimation) can readily been realized with our reported efficiencies. Indeed, all have already been looked at with a smaller number of measurement channels and small efficiencies (much larger inefficiencies than 1%) However, it is clear that they would all benefit from increasing the efficiency values.

3. In Fig. S8b, the direction the arrows are aiming at does not obviously look to be $\sigma_z = -1$. How should I interpret it?

Reply:

We thank the referee for pointing out this issue. Our formulation was quite unfortunate. Indeed, each vector does not point straightly towards the $z=-1$ pole. Instead the vector field corresponds to field lines that all go through the south pole. More specifically, the field lines are along spheroids that are symmetric around z and that go through the pole $z=-1$. This is done in our Figure S4 of [10] in the case of fluorescence alone.

We have changed the caption accordingly and corrected a typo switching u and w .

“The arrows form a vector field whose field lines end up at the pointer states of the measurement namely $\sigma_z = \pm 1$ for the dispersive measurement (Fig.~\textbf{a}) and $\sigma_z = -1$ for the fluorescence measurement (Fig.~\textbf{b}), where no backaction occurs.”

REVIEWERS' COMMENTS:

Reviewer #1 (Remarks to the Author):

Following the authors reply and changes, I can now recommend the paper for publication as is.

Additionally the authors may want to consider citing the following paper:

Single-shot quantum state estimation via a continuous measurement in the strong backaction regime

RL Cook, CA Riofrío, IH Deutsch

Physical Review A 90 (3), 032113

I believe it relates to their tomography discussion